# Numerical simulation of non-neutral forest canopy flows at a site in North-Eastern France

Cian J. Desmond<sup>1</sup>, Simon J. Watson<sup>2</sup>, Christiane Montavon<sup>3</sup>, Jimmy Murphy<sup>1</sup>

<sup>1</sup>MaREI, University college Cork, Ireland. <sup>2</sup>DUWIND, Delft University of Technology <sup>3</sup>ANSYS Europe Ltd

Correspondence to: Simon J. Watson (s.j.watson@tudelft.nl)

- Abstract. The flow over densely forested terrain under neutral and non-neutral conditions is considered using commercially available Computational Fluid Dynamics software. Results are validated against data from a site in North-Eastern France. It is shown that the effects of both neutral and stable atmospheric stratifications can be modelled numerically using state of the art methodologies whilst unstable stratifications are more difficult to simulate accurately. The sensitivity of the numerical model to parameters such as canopy height, canopy density
- and the turbulence modelling constant  $C_{\mu}$  is also assessed.

### **1** Introduction

The computational power required to run full Computational Fluid Dynamics (CFD) simulations on the scale of a typical wind farm is now accessible and as a result, CFD is beginning to see greater adoption by industry for the purposes of wind resource assessment [Mortensen & Jørgensen (2013)]. Following this trend, research activities

have increased into the flow dynamics generated by non-trivial terrain and atmospheric features in order to fully realise the capabilities of CFD to describe the atmospheric boundary layer (ABL) and to meet the demanded uncertainty standards.

One element of terrain complexity which has been found to significantly increase flow modelling uncertainty is the presence of forestry. It was shown in Brower et al. (2014) that forestry increases modelling uncertainty in

- terms of root mean square error by a factor of 4-5 when modelling the flow between mast pairs using a variety of industry standard modelling software packages. In Desmond & Watson (2014) it was suggested that one reason for these elevated levels of uncertainty may be the regular occurrence of non-neutral atmospheric stability events in forested terrain. The buoyancy forces associated with non-neutral events are generally neglected in industry standard modelling software packages, however, they have been shown to have a significant impact on how the
- wind interacts with obstacles such as forestry [Brunet & Irvine (2000), Morse et al. (2002) and Chaudhari et al. (2016)].

In Desmond et al. (2017) the possibility of including the joint effects of atmospheric stability and forest canopy drag within a CFD domain was examined through the use of validation data from stratified ABL wind tunnel experiments. Whilst the results achieved in Desmond et al. (2017) were promising, the analysis was limited by a

35 lack of availability of experimental data for an unstably stratified ABL and also possible Reynolds number scaling problems when using architectural model trees to represent a forest canopy.

For this paper, non-neutral Reynolds Averaged Navier Stokes (RANS) CFD simulations have been validated against field data from a heavily forested site in North-Eastern France. Firstly, sets of stable, neutral and unstable events are identified. The neutral events are then numerically modelled in order to identify the appropriate terrain,

canopy, mesh and atmospheric configurations to successfully model flow over the site. The effects of atmospheric stability are then introduced in an attempt to replicate the non-neutral events observed in the dataset.

All CFD simulations in this paper have been configured using the WindModeller (WM) software package which is a front end for the ANSYS CFX flow solver. WM has been specifically designed to meet the needs of the wind

energy industry and it includes the ability to simulate the effects of non-neutral stability.

In Section 2 of this paper the validation data are presented and in Section 3 the CFD configurations are discussed. Sections 4, 5 and 6 present the validation results and discussion for the neutral, stable and unstable intercomparisons respectively. Section 7 presents overall conclusions.

# 2 Validation data

This study uses data from a mast near Vaudeville which is located adjacent to a wind farm in North-Eastern France (46° 26' 58"N, 05° 35' 02"E). There is an extensive mixed forest located to the west at a distance of c. 170 m as shown in Figure 1.

Figure 1. Location of the Vaudeville met mast is indicated by the red marker. [Picture credit: www.maps.google.com]

An Institut National de l'Information Géographique et Forestière (IGN) map of the area under consideration is given in Figure 2. Four operating turbines are marked on this map; the two closest turbines to the mast are located at a bearing of  $85^{\circ}$  and a distance of 400 m and  $25^{\circ}$  at a distance of 600 m.