# Peer review of "Numerical simulation of non-neutral forest canopy flows at a site in North-Eastern France"

_Wind Energy Science, 2017_

## Referee Comment (RC1) · Anonymous Referee #1 · 25 Sep 2017

Review on the MS wes-2017-34 Numerical simulation of non-neutral forest canopy flows at a site in North-Eastern France by Cian J. Desmond, Simon J. Watson, Christiane Montavon and Jimmy Murphy

The authors try to validate commercially available Computational Fluid Dynamics software (after some modification) against measurements data from a site in North-Eastern France. They conclude that this CFD model is able "to simulate the joint effects of canopy drag and atmospheric stability when considering stable stratification", but unable "to simulate the unstable events in the validation dataset". While I agree with the general idea that in scientific research the negative results could also be quite useful and publishable, the results of this paper do not satisfy these conditions. I have serious reservations about the methodology and model's application to simulate the key

parameters needed for wind risk assessment. The conclusions of the paper are poorly written and don't provide any new results or specific recommendations.

Recommendation

I recommend the paper to be rejected.

Main comments

1. I am not sure that threshold values for wind shear and turbulence intensity chosen for conditions of high wind speeds as indicator of neutral stability of atmosphere are suitable for other conditions. "Narrow range of values for wind shear", which is rather insensitive to solar irradiance, with high probability indicates the convective regime of atmosphere. Simultaneous increase of turbulence intensity with increasing solar irradiance (Fig. 5) also confirms this state of atmosphere. Thus, identification of stable, neutral and unstable events as shown in Fig. 6 is wrong. The authors obviously identified the atmosphere states based on the data from the height of 80 m only, which has nothing in common with real atmospheric stability. It can explain why the model results don't match with identified stability regimes. I do not understand why the authors did not check identified regimes against the measurement data as it has been done in Desmond and Watson (2014) for Norunda site. It seems that the set of sensors used in measurements allows the identification of all parameters needed including heat flux, temperature and wind speed profiles.

2. Description of the model is not sufficient for readers, who do not work regularly with WindModeller software package (the authors didn't provide any references). Thus it is difficult to understand how the model describes the stratified flows, specifically what kind of equations are used? Before applying the model to the real situation, I would advise to test the model against simplified flows over an open place or forest in one-dimensional mode.

3. Without any proof that the model adequately describes main flow properties in
atmospheric boundary layer, the authors were unsuccessfully trying to identify parameters and boundary conditions of model that would provide the better fit with validation dataset. Actually, in conclusions they mentioned, that "due to the fact that validation data is limited to a single measurement location, it will not be possible to fully appreciate the ramifications of such alterations on the overall quality of the simulation". It seems that only this fact did stop them from new numerical experiments.

4. Generally I did not find any clear strategy in modelling experiments – most of them could be performed in one-dimensional mode. For example, I consider that numerical experiments with $C\mu$ value were absolutely superfluous, because $C\mu$ in CFD models is strictly related to TKE, and therefore to Turbulence Intensity defined in the paper by Eq. 12. Playing with vegetation parameters without information on real vegetation looks also weird. On page 20, lines 18-21, the authors came to conclusion that "the average LAD for the Vaudeville forest is approximately 3 m-1 ", which with h = 10 m will provide incredibly dense forest with LAI = 30. I understand that the model can accept any LAD values as well as any surface temperature, but more realistic values would be better.

5. Finally, the paper provides an impression that it was hastily written; there are many imprecise and incomplete formulations and references in the text.

Desmond, C., Watson S., 2014. A study of stability effects in forested terrain. Journal of Physics: Conference Series 555.

---

## Referee Comment (RC2) · Anonymous Referee #2 · 17 Oct 2017

The paper reports results of numerical simulations performed over a forested terrain. Meteorological mast measurements are available for validation purposes of the RANS simulations. The authors describe carefully the measurement equipment and the simulation details, but the result quality is very questionable in several crucial aspects

- I was quite surprised that, despite the fact that the mast has tri-axial sonic anemometers, the authors did not use the fluctuating temperature from the sonics to estimate the Monin-Obukhov length, $L_o$, which is the actual quantity used to discern between stable, unstable and neutral conditions. Another alternative would be the Richardson number, but the uncertainty in the temperature gradient will limit its calculation. There have been several works (see for instance Medici and co-workers at the EWEA conference 2014) that have analysed proxies to get

stratification conditions over forested sites and the problems arise especially on the thresholds to be used to discern between the various states: The authors give some values at page 6 without any clear explanation about their origin. In my opinion, the Obukhov length is the best method to discern between different stability conditions although it is rarely available since many masts have only cup anemometers, vanes and some thermometer.

- The method the authors adopted to get the friction velocity is quite strange and strongly relies on the existence of a logarithmic layer. This is not the way wall functions are introduced, for instance, and the value of $k$ at the wall should be instead used (assuming a Neumann condition for $k$). In case of stable or unstable boundary layer, the $\psi_m(z/L_o)$ function sums to the velocity profile, with an increasing deviation from the logarithmic behaviour, so that their approach is clearly problematic. The fact that they change the reference height were the friction velocity can be estimated (up to 500 m!) indicates that they have little familiarity about what the friction velocity is and the structure of the turbulent boundary layer.

- One of the biggest flaws of the manuscript is that the authors have changed several parameters ($h_c$, $L_x$, $T_{wall}$ and even $C_\mu$ in the turbulence model) to get quantitative match with the experimental data. Driven by the idea that every model can fit experimental data if one varies the parameters enough, their approach is justified, but unfortunately this is unacceptable in science. What if they had to do another evaluation where the true answer is not available? Rather than doing 47 simulations to find the right parameters, they could have just estimate the average forest height from the available measurements, estimate the LAI from publications or reported values and do nothing more.

- Following the previous comment, I find quite funny that the authors decide to simulate a forest that is twice higher than the real one (they use the settings of

simulation 38 for the stable and unstable cases) just because it fits the velocity data. Furthermore, having $L_x = 0.7 m^{-1}$ implies a LAI equal to 70 (according to the estimate of Harman & Finnigan), which is really high. The force is so hight that probably almost no flow is present inside the forest.

- Since many PT-100 were available, the vertical temperature gradient was already known, so that I see no reason to perform the stable and unstable simulations where the floor temperature was changed without any criterion. Simulation 51 for instance uses a temperature decrease of 10 K. Did they observe such a high temperature drop in their experimental data?

- The unstable condition is just inconclusive and counterproductive for the paper. The authors underlined that they could not achieve good results there, so that that section adds nothing to the paper.

- I think that the requirement of more validation data in the conclusions is inappropriate. The reality is that they simply need a better solver or forest model. Once they get acceptable results, they could move to other sites in order to validate their methodology.

**Minor comments**

- The paper from Harman & Finnigan (BLM 2007) should be probably used by the authors. There the authors reported an analysis of the forest boundary layer and proposed a simplified relationship between the loss coefficient $L_x$ and the forest properties as $L_x \approx LAI/(5h_c)$, where $LAI$ is the leaf-area index, $h_c$ is the canopy height and the 5 comes from the assumption of $c_d \approx 0.2$. Usually, a LAI between 1 and 4 is observed, so that $L_x$ should be here around 0.04, namely the standard value proposed by WM.

[Figure]

- The interesting paper from Silva Lopes et al. (BLM 2013) with title *Improving a Two-Equation Turbulence Model for Canopy Flows Using Large-Eddy Simulation* could provide some suggestions to the authors about how to better account for forestry in the $k$ and $\epsilon$ equations.

- The comparison shown in figure 8 is unfair as the image on the right has all tree heights there. Besides, the range 2-5 m is not even around the average tree height

- The authors mention that it is possible to alter the temperature at the ground to introduce stratification effects. Are they using a code with the Boussinesq approximation?

---

## Author Comment (AC1) · 5 Feb 2018

Thank you for taking the time to review our paper in such detail. I have included the revised version and a reply to your comments as a supplement.

I hope that you can download these and that they address your comments to your satisfaction, we have added considerable detail on the numerical formulation as requested.

Please also note the supplement to this comment:
https://www.wind-energ-sci-discuss.net/wes-2017-34/wes-2017-34-AC1-supplement.zip